# Most ReLU Networks Suffer from $\ell^2$ Adversarial Perturbations

**Amit Daniely**
School of Computer Science and Engineering, The Hebrew University, Jerusalem, Israel and Google Research Tel-Aviv
`amit.daniely@mail.huji.ac.il`

**Hadas Schacham**
School of Computer Science and Engineering, The Hebrew University, Jerusalem, Israel
`hadas.schacham@mail.huji.ac.il`

## Abstract

We consider ReLU networks with random weights, in which the dimension decreases at each layer. We show that for most such networks, most examples $x$ admit an adversarial perturbation at an Euclidean distance of $O\left(\frac{\|x\|}{\sqrt{d}}\right)$, where $d$ is the input dimension. Moreover, this perturbation can be found via gradient flow, as well as gradient descent with sufficiently small steps. This result can be seen as an explanation to the abundance of adversarial examples, and to the fact that they are found via gradient descent.

## 1 Introduction

Since the seminal paper of Szegedy et al. [17], adversarial examples arose much attention in machine learning, with various attacks (e.g. [1, 4, 5, 9, 10]) and defence methods (e.g. [12, 13, 11, 20, 7]) being developed, as well as various attempts to explain their presence (e.g. [6, 15, 16, 14, 3]). Yet, it is still not clear why adversarial examples exist, and why they can be found via simple algorithms such as gradient descent.

In this paper we shed new light on the source of this phenomenon, and show that for certain network architectures, for most choices of weights and for most examples $x$, an adversarial example at a Euclidean distance of $\tilde{O}\left(\frac{\|x\|}{\sqrt{d}}\right)$ is guaranteed to exists. Specifically, we show that this holds if each layer reduces the dimension.

Moreover, we show that gradient flow (a continuous analog of gradient descent), or gradient descent with sufficiently small steps, is guaranteed to find these adversarial examples. This result demonstrates that unless we impose restrictions on the weights and/or the examples, we should expect the phenomenon of adversarial examples to occur.

To the best of our knowledge, this is the first result which shows existence of adversarial examples w.r.t. the Euclidean distance for a large class of networks and distributions. Likewise, it is the first result that shows that gradient based algorithms are guaranteed to find such perturbations.

### 1.1 Related Work

Several recent theoretical papers have addressed the question of why adversarial examples exist in machine learning. Schmidt et al. [14] show that the sample complexity of training adversarially robust classifiers might be larger than standard training, while Bubeck et al. [3] show other cases, in which adversarially robust training is computationally harder than standard training.

Fawzi et al. [6] use concentration of measure result to show that for several subsets of $\mathbb{R}^d$, such as the sphere, the ball, or the cube, any partition of the set into a few subset of non-negligible mass (w.r.t. to the uniform measure on these spaces) will result with abundance of adversarial examples. That is, most examples will have a nearby example that belongs to a different part of the partition. This result shows that *any* classifier that realizes this partition will suffer form adversarial examples. Shafahi et al. [15] extend these results to classification tasks in which the examples are generated by certain generative models.

As opposed to our result, in the results of [6, 15] the existence of adversarial examples leans on the input distribution rather than the network which is used for classification. That is, their result show that in the cases under study *any* classifier will suffer from adversarial perturbation. This is in contrast to our result (e.g. corollary 3.1 which applies to any input distribution) in which the existence of adversarial perturbation leans on the network. In other words, we show existence of adversarial examples even in cases where there exists some classifier that does not suffer from adversarial examples.

We argue that in the context of adversarial examples, cases in which there is some classifier that does not suffer from adversarial examples are of particular interest. Indeed, the very existence of adversarial examples hinges on the fact that humans solves the task at hand without suffering from adversarial examples.

Lastly, let us mention Shamir et al. [16] which is the closest to our work, and in fact inspired this paper. They proved that for ReLU networks, under rather mild conditions, *any example* will have an adversarial perturbation with small $\ell_0$ distance. That is, it is possible to change just a few input coordinates to generate an adversarial perturbation. Moreover, they have shown that such a perturbation can be found by a simplex-like algorithm.

The advantage of Shamir et al. [16] is that the conditions on the network's weights are rather mild. On the other hand, the advantage of our approach is that we consider the $\ell_2$ distance which is more natural than the $\ell_0$ distance. Indeed, in Shamir et al. [16] there is no bound on the magnitude of the movement that is required in every coordinate. In particular, the guaranteed adversarial perturbation may have coordinates that are out of the relevant range, and therefore can be detected easily or even won't be deemed as a legal input to the network. The following example demonstrate that this might happen even in very simple settings.

**Example 1.** *Consider any example $x$ in $[-1, 1]^d$, and a linear classifier $w \in \left[ -\frac{1}{\sqrt{d}}, \frac{1}{\sqrt{d}} \right]^d$ such that $w^\top x \geq 1$. As explained in section 2, the scale of $x$, $w$ and $w^\top x$ is rather standard. Suppose that we want to find an adversarial perturbation to $x$ by changing just $O(1)$ coordinates. It is not hard to see that the magnitude of the change in at least one of the coordinates must be $\Omega(\sqrt{d})$. As the original input $x$ is in $[-1, 1]^d$ it is quite likely that the range of the adversarial perturbation won't result in a reasonable input.*

Another advantage of our result is that in contrast to Shamir et al. [16] who consider a simplex-like algorithm for finding adversarial perturbation, we use gradient flow (or gradient descent with sufficiently small steps). As adversarial examples are usually sought with gradient based algorithms, our result gives a better explanation to why adversarial perturbations are found in practice.

## 2 Preliminaries

**Neural Networks** We will consider fully connected ReLU neural networks defined by weights $\vec{W} = (W_1, \ldots, W_{t-1}, W_t)$, where for each $1 \leq j \leq t$ $W_j$ is a $d_{j+1} \times d_j$ matrix. We denote the input dimension by $d = d_1$ and assume that the output dimension $d_{t+1}$ is 1. The function computed by the network defined by the weights $\vec{W}$ is

$$h_{\vec{W}}(x) = W_t \circ \sigma \circ W_{t-1} \circ \ldots \circ \sigma \circ W_1(x)$$

where $\sigma$ is the ReLU function, with the convention that when it is applied on vectors it operates coordinate-wise. We will denote by $h_{W_1, \ldots, W_i}(x)$ the output of the $i$'th layer, before the ReLU is applied, and by $\sigma(h_{W_1, \ldots, W_i}(x))$ the output of the $i$'th layer after the ReLU is applied.

**Random Weights** We next describe the distribution over the space of weights that we will consider. A *random weight matrix* is a $k \times d$ matrix whose elements are i.i.d. centered Gaussians. *Random*

*weights* are weights $\vec{W} = (W_1, \ldots, W_{t-1}, W_t)$ where for each $1 \le j \le t$, $W_j$ is a random weight matrix. We say that a random $k \times d$ weight matrix in *normalized* if the variance of the Gaussians is $\frac{1}{d}$. This normalization is rather standard [8] in both theory and practice of neural networks, as under it the scale of the weights resembles the scale of weights in real world networks. Indeed, for a fixed example whose coordinates have magnitude of $O(1)$, the magnitude (the second moment to be precise) of the input to all neurons is $O(1)$ (see [8]).

**Gradient flow**  Given a function $h : \mathbb{R}^d \to \mathbb{R}$ and a point $x_0 \in \mathbb{R}^d$, the gradient flow starting at $x_0$ is the trajectory $\gamma(t)$ that satisfies $\gamma(0) = x_0$ and $\gamma'(t) = -\nabla h(\gamma(t))$. We note that gradient descent is a discretization of gradient flow, and its trajectory becomes closer and closer to the trajectory of gradient flow as the step size gets closer to 0. In the context of adversarial examples, given weights $\vec{W}$ and an example $x_0 \in \mathbb{R}^d$, adversarial perturbation is often sought by performing gradient flow over the function $x \mapsto y h_{\vec{W}}(x)$, where $y = \mathrm{sign}(h_{\vec{W}}(x_0))$.

**Convention**  Throughout the paper, big-O notations are w.r.t. the input dimension $d$.

## 3  Results

### 3.1  Result for Random Matrices

Our first result considers networks in which the dimension decreases in every layer. It shows that for most such networks, most examples $x$ will have an example $x'$ such that (1) its distance from $x$ is $\tilde{O}\left(\frac{\|x\|}{\sqrt{d}}\right)$ and (2) $\mathrm{sign}(h_{\vec{W}}(x')) \ne \mathrm{sign}(h_{\vec{W}}(x))$. Moreover, $x'$ can be found by gradient flow.

**Theorem 3.1.** *Assume that for any $1 \le j \le t$, $d_{j+1} = o(d_j)$ and that $d_t = \omega(1)$. Fix any non-zero example $x_0 \in \mathbb{R}^d$ and let $\vec{W}$ be random weights. Then, w.p. $1 - o(1)$, gradient flow of length $\tilde{O}\left(\frac{\|x_0\|}{\sqrt{d}}\right)$ starting at $x_0$ will flip the sign of network's output.*

An immediate implication of theorem 3.1 together with Markov's inequality is that given *any* input distribution $\mathcal{D}$ and a random network, most examples according to $\mathcal{D}$ will have a close adversarial perturbation, which can be found by gradient flow:

**Corollary 3.1.** *Assume that for any $1 \le j \le t$, $d_{j+1} = o(d_j)$ and that $d_t = \omega(\ln(d))$. Fix a distribution $\mathcal{D}$ on $\mathbb{R}^d$ and let $\vec{W}$ be random weights. Then, w.p. $1 - o(1)$ over the choice of $\vec{W}$ the following will hold. If $x_0 \sim \mathcal{D}$, then w.p. $1 - o(1)$ over the choice of $x_0$ gradient flow of length $\tilde{O}\left(\frac{\|x_0\|}{\sqrt{d}}\right)$ starting at $x_0$ will flip the sign of network's output.*

### 3.2  Result for Strongly Surjective Matrices

To prove theorem 3.1 we show that (1) random matrices that reduce the dimension have strong-surjectivity properties, and that (2) for any network whose matrices possess this surjectivity property, any "typical" example has a close adversarial perturbation that can be found via gradient flow. As we find the second result of independent interest, we outline it next.

To this end, we first define and motivate the aforementioned surjectivity property. For simplicity, we will work with normalized weights. We note that due to the homogeneity of the ReLU, our results for random weights are insensitive to the variance of the weights. Hence, restricting to normalized weights does not limit the generality of the result.

We denote by $\mathbb{B}^d$ the unit ball in $\mathbb{R}^d$. For a constant $c > 0$, we say that a $k \times d$ matrix $W$ is *c-surjective* if $c\mathbb{B}^k \subset W\mathbb{B}^d$. We note that if $W$ is a normalized random weight matrix with $d = (1 + \Omega(1))k$ then $W$ is $\Omega(1)$-surjective[1] w.h.p. We will rely on a stronger surjectivity property that is still valid w.h.p. We say that $W$ is $(c_1, c_2)$-surjective if any matrix that is composed of $\ge c_1 d$ columns from $W$ is $c_2$-surjective. The following result shows that if $k = o(d)$ (as in the case of theorem 3.1), and $W$ is a random weight matrix, then for any constant $c_1 > 0$, $W$ is $(c_1, \Omega(1))$-surjective w.h.p.

**Theorem 3.2.** *Fix a constant $1 > c_1 > 0$. There are constants $c_2, c_3 > 0$, that depend only on $c_1$, for which the following holds. Let $W$ be a normalized random $k \times d$ weight matrix with $k \leq c_3 d$. Then, w.p. $1 - 2^{-\Omega(d)}$, $W$ is $(c_1, c_2)$-surjective.*

In light of theorem 3.2 we say that weights $\vec{W} = (W_1, \ldots, W_{t-1}, W_t)$ are $(c_1, c_2)$-*typical* if for every $t$ the matrix $W_t$ is $(c_1, c_2)$-surjective and has spectral norm at most $\frac{1}{c_2}$. Theorem 3.2 together with theorem 4.1 implies that for any constant $c_1 > 0$, if $\vec{W}$ are normalized random weights with dimensions as in theorem 3.1, then they are $(c_1, \Omega(1))$-typical w.h.p. We say that an example is $(c_1, c_2)$-*typical w.r.t.* $\vec{W}$ if (1) in every layer the input value of at least $2c_1$ fraction of the neurons is $\geq \frac{\|x\| \cdot c_2}{\sqrt{d}}$ and (2) $|h_{\vec{W}}(x)| \leq \|x\| \sqrt{\frac{\ln(d)}{d}}$. It is holds that for any constant $c_1 < \frac{1}{4}$, and for any example $x$, if $\vec{W}$ are normalized random weights then w.h.p. over the choice of $\vec{W}$, the example $x$ is $(c_1, \Omega(1))$-typical w.r.t. $\vec{W}$. (see lemma 4.4)

**Theorem 3.3.** *Fix constants $1 > c_1, c_2 > 0$ and depth $t \in \{2, 3, \ldots\}$. Assume that $d_t = \omega(\ln(d))$, that $\vec{W}$ are $(c_1, c_2)$-typical weights, and that $x_0$ is a $(c_1, c_2)$-typical example w.r.t. $\vec{W}$. Then, gradient flow of length $\tilde{O}\left(\frac{\|x_0\|}{\sqrt{d}}\right)$ starting at $x_0$ will flip the sign of the network's output.*

# 4 Proofs

## 4.1 Preliminaries and Notation

We denote by $\mathbb{S}^{d-1}$ the unit sphere in $\mathbb{R}^d$. For $x \in \mathbb{R}^d$ and $r > 0$ we denote by $B(x, r)$ the closed ball of radius $r$ around $x$. An $\epsilon$-cover of a set $A \subset \mathbb{R}^d$ is a set $S \subset A$ such that for any $x \in A$ there is $y \in S$ with $\|x - y\| < \epsilon$. We will use the following well known result for random Gaussian matrices:

**Theorem 4.1** (E.g. Corollary 5.35 in [19]). *Suppose that $W$ is a random $n \times m$ matrix with i.i.d. Gaussian entries of mean $0$ and variance $\frac{1}{d}$. Then, for every $t > 0$, w.p. at least $1 - 2\exp\left(-\frac{dt^2}{2}\right)$*

$$\frac{\sqrt{m} - \sqrt{n}}{\sqrt{d}} - t \leq s_{\min}(W) \leq s_{\max}(W) \leq \frac{\sqrt{m} + \sqrt{n}}{\sqrt{d}} + t$$

We will also use the following separation theorem for convex sets:

**Theorem 4.2** (E.g. Chapter 2 in [2]). *Let $C \subset \mathbb{R}^d$ be a closed and convex set, and let $x \in \mathbb{R}^d \setminus C$. There is a vector $u \in \mathbb{S}^{d-1}$ such that $\sup_{y \in C} u^\top y < u^\top x$*

## 4.2 Proof of Theorem 3.2

Let $W$ be a $k \times d$ matrix. For $A \subset [d]$ we denote by $W^A$ the matrix that is obtained from $W$ upon zeroing all entries $w_{ij}$ with $j \notin A$. Note that $W$ is $(c_1, c_2)$-*surjective* if and only if for every $A \subset [d]$ with $|A| \geq c_1 d$, the matrix $W^A$ is $c_2$-surjective.

**Lemma 4.1.** *Fix a constant $1 > c_1 > 0$. There is a positive constant $c_2' = c_2'(c_1)$ for which the following holds. Let $W$ be a random $k \times d$ weight matrix and let $y \in \mathbb{S}^{k-1}$. Then, w.p. $1 - 2^{-\Omega(d)}$ for every set $A$ of size at least $c_1 d$ there is a vector $x \in \mathbb{S}^{d-1}$ such that $y^\top W^A x \geq c_2'$.*

*Proof.* We note that the vector $y^\top W$ is a vector of $d$ independent and centered Gaussians, of variance $\frac{1}{\sqrt{d}}$. Hence, lemma 4.2 below implies that for sufficiently small constant $c_2' > 0$, it holds that w.p. $1 - 2^{-\Omega(d)}$ the sum of the squares of the smallest $\lceil c_1 d \rceil$ elements in $y^\top W$ is at least $c_2'^2$. In this case, for every set $A$ of size at least $c_1 d$, $\|y^\top W^A\| \geq c_2'$, which implies that there is a vector $x \in \mathbb{S}^{d-1}$ such that $y^\top W^A x \geq c_2'$. $\qquad\square$

**Lemma 4.2.** *Fix a constant $c_1 > 0$. There is a constant $c_2 > 0$ for which the following holds. Let $X_1, \ldots, X_d$ i.i.d. standard Gaussian and denote by $Z$ the sum of the smallest $\lceil c_1 d \rceil$ elements in $X_1^2, \ldots, X_d^2$. Then $\Pr(Z > c_2 d) = 1 - 2^{-\Omega(d)}$*

*Proof.* Let $1 > p > 0$ be big enough such that if $Y_1, \ldots, Y_d$ are i.i.d. Bernoulli r.v. with parameter $p$, then $\Pr\left(\sum_{i=1}^d Y_i > \left(1 - \frac{c_1}{2}\right)d\right) = 1 - 2^{-\Omega(d)}$. Such $p$ exists by, say, Hoeffding's inequality. Let $c_2' > 0$ be small enough such that $\Pr_{X \sim \mathcal{N}(0,1)}\left(X^2 > c_2'\right) > p$. By the choice of $c_2'$ and $p$ it holds that w.p. $1 - 2^{-\Omega(d)}$ over the choice of $X_1, \ldots, X_d$ we have that $X_i^2 > c_2'$ for more than $\left(1 - \frac{c_1}{2}\right)d$ $i$'s. In this case, amongst the smallest $\lceil c_1 d \rceil$ elements in $X_1^2, \ldots, X_d^2$, there are at least $\frac{c_1}{2}d$ elements with $X_i^2 > c_2'$, in which case $Z > \frac{c_1 c_2'}{2}d$. All in all, we have shown that for $c_2 = \frac{c_1 c_2'}{2}$, $\Pr\left(Z > c_2 d\right) = 1 - 2^{-\Omega(d)}$ □

Via a union bound and the fact that $\mathbb{S}^{k-1}$ has an $\epsilon$-cover of size $(1/\epsilon)^{O(k)}$ (e.g. chapter 5 in [18]) we conclude that:

**Corollary 4.1.** *Fix a constant $1 > c_1 > 0$ and let $c_2' = c_2'(c_1)$ be the constant from lemma 4.1. Let $S$ be a $\frac{c_2'}{7}$-cover of $\mathbb{S}^{k-1}$ of size $2^{O(k)}$. There is a positive constant $c_3 = c_3(c_1)$ for which the following holds. Let $W$ be a random $k \times d$ weight matrix with $k \leq c_3 d$. Then, w.p. $1 - 2^{-\Omega(d)}$, for every set $A$ of size at least $c_1 d$, and for every $y \in S$, there is a vector $x \in \mathbb{S}^{d-1}$ such that $y^\top W^A x \geq c_2'$.*

**Lemma 4.3.** *Let $S$ be a $\frac{c_2'}{7}$-cover of $\mathbb{S}^{k-1}$, and let $C \subset \mathbb{R}^k$ be a closed and convex set such that (1) for any $y \in S$ there is $x \in C$ such that $y^\top x \geq c_2'$, and (2) $C$ is contained in the ball of radius $3$. Then $C$ contains the ball of radius $\frac{c_2'}{2}$ around zero.*

*Proof.* Assume toward a contradiction that there is a vector $x_0$ with $\|x_0\| \leq \frac{c_2'}{2}$ such that $x_0 \notin C$. By the separation theorem for convex sets (theorem 4.2), there is a unit vector $u \in \mathbb{S}^{d-1}$ such that for any $x \in C$

$$u^\top x < u^\top x_0 \leq \|u\| \cdot \|x_0\| \leq \frac{c_2'}{2} \tag{1}$$

Now, choose $y \in S$ that satisfies $\|y - u\| \leq \frac{c_2'}{7}$, as well as $x \in C$ such that $y^\top x \geq c_2'$. We have

$$y^\top x = u^\top x + (y - u)^\top x \leq \frac{c_2'}{2} + \|y - u\| \cdot \|x\| \leq \frac{c_2'}{2} + \frac{c_2'}{7} \cdot 3 < c_2'$$

contradicting the assumption that $y^\top x \geq c_2$ □

*Proof.* (of theorem 3.2) Let $c_2' = c_2'(c_1)$ and $c_3 = c_3(c_1)$ be the constants from corollary 4.1. Define $c_2 = \frac{c_2'}{2}$. Let $S$ be an $\frac{c_2'}{7}$-cover of $\mathbb{S}^{k-1}$. Let $W$ be a random $k \times d$ weight matrix with $k \leq c_3 d$. By corollary 4.1 and theorem 4.1 we have that w.p. $1 - 2^{-\Omega(d)}$:

1. For every set $A \subset [d]$ of size $\geq c_1 d$ and every $y \in S$, there is $x \in W^A \mathbb{B}^d$ with $y^\top x \geq c_2'$

2. $W^A \mathbb{B}^d$ is contained in the ball of radius $3$ around $0$

Lemma 4.3 implies that $W^A \mathbb{B}^d$ contains the ball of radius $c_2$ around $0$, and hence $W^A$ is $c_2$-surjective. As this is true for any $A \subset [d]$ of size $\geq c_1 d$, it follows that $W$ is $(c_1, c_2)$-surjective. □

### 4.3 Proof of theorem 3.3

W.l.o.g. we assume that $\|x_0\| = \sqrt{d}$. Let $W_i^x$ be the matrix obtained form $W_i$ by replacing each column $j$ corresponding to a neuron that is "off" (that is, their value is 0) with 0. We have that

$$h_{\vec{W}}(x) = W_t^x \cdot W_{t-1}^x \cdot \ldots \cdot W_1^x x$$

hence, the gradient of $h$ at $x$ is $W_t^x \cdot W_{t-1}^x \cdot \ldots \cdot W_1^x$.

Now, fix $x \in B(x_0, c_2^{-t}\sqrt{\ln(d)})$. Since each layer computes a function which is $O(1)$-Lipschitz (as the spectral norm of the weight matrices is $O(1)$), and since there are $O(1)$ layers, we have that the norm of the input vector for each layer changes by at most $O(\sqrt{\ln(d)})$ when moving from $x_0$ to $x$. In particular, at most $O(\log(d))$ neurons whose input value is $\geq c_2$ for $x_0$, become inactive when we move to $x$. Hence, the number of active neurons for $x$ at layer $i$ is at least $2c_1 d_i - O(\ln(d))$, which

is more than $c_1 d_i$ as $d_t = \omega(\ln(d))$ (note that $d_i \geq d_t$. Indeed, since the matrices are surjective, we have that $d_t \leq d_{t-1} \leq \ldots \leq d_1$).

Since the weight matrices are $(c_1, c_2)$-surjective, we have that $W_i^x$ is $c_2$-surjective for any $x \in B(x_0, c_2^{-t} \sqrt{\ln(d)})$. As the composition of $t$ $c_2$-surjective matrices is $c_2^t$ surjective, we have that the gradient $W_t^x \cdot W_{t-1}^x \cdot \ldots \cdot W_1^x$ is $c_2^t$-surjective. As the gradient is a vector, this means that[2] $\|\nabla h_{\vec{W}}(x)\| \geq c_2^t$.

All in all, we have shown that for any $x \in B(x_0, c_2^{-t} \sqrt{\ln(d)})$, the gradient of $h_{\vec{W}}$ at $x$ has norm at least $c_2^t$. Assuming that $\text{sign}(h_{\vec{W}}(x)) = 1$ (respectively, $\text{sign}(h_{\vec{W}}(x)) = -1$), this implies that gradient flow starting at $x_0$ for length of $c_2^{-t} \sqrt{\ln(d)}$ will decrease (respectively, increase) the output of the network by at least $\sqrt{\ln(d)}$, which means that the output will change its sign.

### 4.4  Proof of theorem 3.1

**Lemma 4.4.** *Assume that $d_t = \omega(1)$. For any constant $c_1 < \frac{1}{4}$ and for any example $x \in \mathbb{R}^d$, if $\vec{W}$ are normalized random weights then w.h.p. over the choice of $\vec{W}$, $x$ is $(c_1, \Omega(1))$-typical w.r.t. $\vec{W}$*

*Proof.* (sketch) W.l.o.g. we assume that $\|x\| = \sqrt{d}$. By standard concentration results it holds that w.p. $1 - o(1)$ we have that $\|\sigma(h_{W_1,\ldots,W_{i-1}}(x))\| = \Theta(1)$. Now, given weight matrices $W_1, \ldots, W_{i-1}$ such that $\|\sigma(h_{W_1,\ldots,W_{i-1}}(x))\| = \Theta(1)$, we have that $h_{W_1,\ldots,W_i}(x)$ is a vector of i.i.d centered Gaussians of variance $\Theta(1)$. Hence, by standard concentration results we have that[3] w.p. $1 - 2^{-\Omega(d_{i+1})} = 1 - o(1)$ over the choice of $W_i$, the value of $2c_1$ fraction of the coordinates in $h_{W_1,\ldots,W_i}(x)$ is $\Omega(1)$. Similarly, given $W_1, \ldots, W_{t-1}$ such that $\|\sigma(h_{W_1,\ldots,W_{t-1}}(x))\| = \Theta(1)$, we have that $h_{\vec{W}}(x)$ is a centered Gaussian of variance $\Theta(1)$. Hence, w.p. $1 - o(1)$, $|h_{\vec{W}}(x)| \leq \sqrt{\log(d)}$ $\qquad \square$

*Proof.* (of theorem 3.1) W.l.o.g. we assume that $\|x_0\| = \sqrt{d}$. By lemma 4.4 and theorem 3.2 we have that w.p. $1 - o(1)$ the weights $\vec{W}$ are $(1/5, \Omega(1))$ typical, and that $x_0$ is $(1/5, \Omega(1))$ typical w.r.t $\vec{W}$. Theorem 3.3 therefore implies that w.p. $1 - o(1)$ gradient flow of length $O\left(\sqrt{\log(d)}\right)$ will flip the sign of the network. $\qquad \square$

## 5  An Experiment

We made a small experiment on the MNIST data set (see `https://github.com/hadasdas/L2AdversarialPerturbations`). We normalized the examples to have a norm of $\sqrt{784}$ (784 is the dimension of the examples), and trained networks of depth 2-8, with 100 neurons at every hidden layer. We modified the classification task so that the network was trained to distinguish even from odd digits. We then sampled 1000 examples and sought adversarial example for each of them using GD. Figure 5 shows the histogram and average of the distances in which the adversarial examples were found.

Note that in this settings, $\frac{1}{\sqrt{\|x\|}} = 1$. As Figure 5 demonstrates, for most examples we were able to find an adversarial perturbation at a distance of a few units.

## 6  Open Question

A natural open question is to extend our results to more architectures. In this regard we conjecture that theorem 3.1 remains valid without the assumption that $d_{i+1} = o(d_i)$. We also conjecture that an analogous result is valid for convolutional networks.

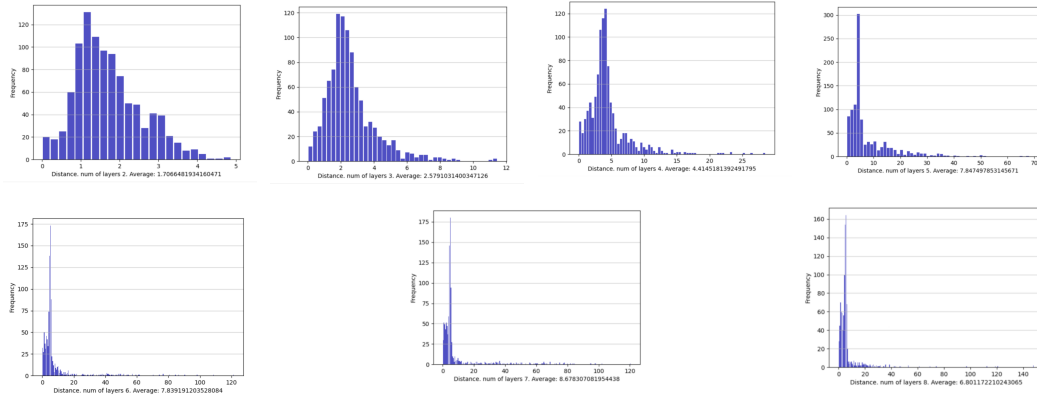

Figure 1: Distance Histograms

## Broader Impact

Not applicable as far as we can see (this is a purely theoretical paper).

## Acknowledgments and Disclosure of Funding

This research is partially supported by ISF grant 2258/19

## Footnotes

[1] This is implied by theorem 4.1 below, together with the fact that $W$ is $c$-surjective if and only if its least singular value is $\ge c$.

[2]Note that a vector $x$ is $c$-surjective if and only if $\|x\| \geq c$.

[3]Note again that the assumption that $d_t = \omega(1)$ implies that $d_i = \omega(1)$ for all $i$.

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
