[Reviews · NeurIPS 2020]

Review 1

Summary and Contributions: Shows that a random fully connected net with random edge values suffers from adversarial examples (for most inputs, whp). The adversarial perturbation needed has small l_2 norm. Furthermore, simple gradient descent can find the adversarial perturbations needed to flip the output. The authors note this can be seen as a followup to Shamir et 2019, which had a result about adversarial perturbations in l_\infty norm. ps Thanks for the rebuttal.

Strengths: (a) The result is simple but nice to have it proven. When I read the abstract it sounded possibly trivial. My guess of the proof strategy was roughly right ---the calculations seems similar to those in recent works on neural tangent kernels and overparametrized nets. But there was some novelty, especially proving that gradient descent finds the perturbations. So I think it is a reasonable result. (b) Fairly short; the paper is only 6 pages long.

Weaknesses: (a) Works only for fully connected nets (b) Requires the width of the layers to decrease monotonically (actually by more than a constant factor) (c) Involves a "constant" which is exponential in depth. (d) Due to (c) it would have been interesting to know if experimentally if the proved lemmas hold for realistic net sizes. In fact this experiment is simple enough to run that authors should clarify this in the rebuttal.

Correctness: Seems fine. No empirical verification (see weakness (d))

Clarity: OK. Could have been fleshed out a bit more for readers who don't have the technical background. A shame because the paper is well under the page limit.

Relation to Prior Work: Yes

Reproducibility: Yes

Additional Feedback:


Review 2

Summary and Contributions: The authors use random matrix theory (RMT) and gradient-flow techniques to show that with high probability over ReLU networks with Gaussian iid weights (properly normalized so that variance ~ 1 / d) and non-increasing layer widths, every example x has adversarial examples of size O(||x||/sqrt(d)). Here d is the dimensionality of the input.

Strengths: - The conceptual contribution of the paper is significant. It prsents a completely new way to understand the phenomenon of adversarial examples, via the lens of RMT and gradient-flow. - The main idea is to show that the gradient of the neural-net y = sign(h(x)) w.r.t the input x is bounded away from zero w.h.p. This in turn implies that, start at each point x, gradient flow will cross the decision boundary after a small time interval. - I've learnt something new reading the paper.

Weaknesses: Minor ------- - There are some issues with the small-/big-Oh notation used in the paper. The statements between line 121 and 125 highlight this confusion. Indeed, k = o(d) Should normally mean lim_{d -> infty} k / d = 0. However, the author seems to be using it in the sense that the exists c > 0 such that k \le c*d for sufficiently large d. The should have been written d' = Omega(d) instead. To be on the safe side, the authors should clearly state what they mean by big-Oh, small-Oh, Omega, etc. This should would only take a 2 lines of the manuscript. - The paper is missing empirical insight. It shouldn't be difficult to run a few experiments on say an MLP neural net on MNIST data and report some curves which consolidate the theoretical claims. There is space in the manuscript to include such material. Insufficient "Related works" ---------------------------- My main issue with this paper is the incompleteness of the outlined "related works". We now have pretty mature picture on why adversarial examples exist (under different geometric conditions on the classifier and / or the distribution of the data). The related works section doesn't do justice to this rich literature (except 1 or 2 relevant references mentioned in the manuscript...). If every new paper starts with a punch-line like "Adversarial examples are mysterious. In this paper we will proof a powerful theorem which explains why they exist." (without mentioning existing literature in this direction), the field will not go very far. Here is a small sample of the existing literature. * Gilmer et al. "The Relationship Between High-Dimensional Geometry and Adversarial Examples" (2018) * Dohmatob, "Generalized No Free Lunch theorem for adversarial robustness" (ICML 2019) * Mahloujifar et al., "The curse of concentration in robust learning" (AAAI 2019) In the above papers, adversarial examples exist as a consequence of ordinary test-error in high-dimensional problems with concentrated class-conditional distributions. Concerning universal lower bounds on adversarial robustness error (i.e of any classifier), one should also mention the recent work: * Bhagoji et al. "Lower bounds on adversarial robustness from optimal transport" (NeurIPS 2019).

Correctness: - The paper is well-rewritten and easy to follow. - The theoretical claims are sound and the proofs seem correct.

Clarity: Yes, the paper is well-rewritten and easy to follow.

Relation to Prior Work: Related works ------------------ I doin't think the outlined "related works" is sufficient. We now have pretty mature picture on why adversarial examples exist (under different geometric conditions on the classifier and / or the distribution of the data). The related works section doesn't do justice to this rich literature (except 1 or 2 relevant references mentioned in the manuscript...). If every new paper starts with a punch-line like "Adversarial examples are mysterious. In this paper we will proof a powerful theorem which explains why they exist." (without mentioning existing literature in this direction), the field will not go very far. Here is a small sample of the existing literature. * Gilmer et al. "The Relationship Between High-Dimensional Geometry and Adversarial Examples" (2018) * Dohmatob, "Generalized No Free Lunch theorem for adversarial robustness" (ICML 2019) * Mahloujifar et al., "The curse of concentration in robust learning" (AAAI 2019) In the above papers, adversarial examples exist as a consequence of ordinary test-error in high-dimensional problems with concentrated class-conditional distributions. - Concerning universal lower bounds on adversarial robustness error (i.e of any classifier), one should also mention the recent work: * Bhagoji et al. "Lower bounds on adversarial robustness from optimal transport" (NeurIPS 2019). Add more material ? ---------------------------- - The manuscript is really short (6 pages, but almost every NeurIPS paper is 8 pages long, as permitted by the guidelines). There is room for more material. For example, a few plots showing the geometric intuition behind the approach would help the general reader (i.e the general reader shouldn't be required to known the key results from RMT in order to appreciate the paper).

Reproducibility: Yes

Additional Feedback:


Review 3

Summary and Contributions: This paper demonstrates that for random ReLU networks with binary outputs with decreasing dimension per layer, for most such networks, for most inputs x, one can flip the output for input x by simply running gradient flow to find an adversarial perturbation to x which is bounded l2 distance away from x.

Strengths: It is interesting to demonstrate that it is easy to find adversarial examples in l2 distance using gradient flow/descent for a random network. This paper gives some theory for this practically observed phenomenon. Indeed, this paper answers an open problem from https://arxiv.org/pdf/1901.10861.pdf.

Weaknesses: The scope is somewhat narrow, but I think this is fine for a theory paper. Additionally, perhaps a more thorough paper would also answer whether real, trained networks are similar to random networks in this manner -- when and when not? But this is perhaps a question for future research. === AFTER RESPONSE ==== Basically my only additional suggestion is that the full 8 pages should be used (see my other suggestions, and the suggestions from the other reviews).

Correctness: I followed many of the proofs, but not all of them. I think it would be helpful if a few more details were provided and the proofs were re-written with more clarity and high-level guideposting (e.g., you can say that the main idea is to lower bound the gradient). The results seem ok, but maybe another reviewer checked in more detail. Also Lemma 4.4. should be given in detail.

Clarity: There are a few issues with clarity in the introduction. It should be clarified what is meant by an "example x" -- I think what is meant is that for input x and output f(x), there is a small perturbation at x such that f(x) is different -- but there should be more clarity about "f(x) is different" -- what is being assumed about the network output? These are addressed later on, but it is confusing to read in the introduction. There are a few other places where wording is confusing, see Additional Remarks. I also thought the proof could have used better organization (there's a lot of jumping around between results out of order) and more of an outline/ guideposts for the reader; it was overly dense as is. In particular it would be nice to say upfront why surjectivity is important and what the intuition regarding it is -- e.g., why does surjectivity help for proving gradient flow leads to perturbed adversarial examples in line 110? Because you want to lower bound the gradient in terms of the typicalness constant and use the Lipschitz property of the network to argue the perturbation occurs.

Relation to Prior Work: Yes, sufficient discussion of prior work is in the paper.

Reproducibility: Yes

Additional Feedback: Typo at line 16: "exist" instead of "exists". similar at line 42. Line 32: "will result with abundance" is not correct grammar -- "result in an abundance" line 34: "form" --> "from" I found lines 41-47 confusingly worded -- you show that there can be adversarial examples for random networks even if, for the input distribution, there exists some classifier which does not have adversarial examples; as opposed to a setting where by nature of the distribution, every classifier has an adversarial example. Thus it is a property of the randomness of the network that there is an adversarial example. For some reason it took a while to parse this, maybe re-word a bit? Similarly, line 58: "demonstrates" instead of 'demonstrate' I found Example 1 a bit confusingly worded as well -- you are just saying that l-infty balls for adversarial perturbations are much larger than l2 balls for adversarial perturbations, right? Maybe restate in that sort of language. In the Results, should probably mention that the gradient flow is being optimized with a loss to flip h explicitly and precisely. Please include 117-122 in a definition rather than just plain text. Do you define s_min and s_max in Thm 4.1.? Are these singular values? It seems so based on the footnote, but these should really be defined. Better organization needed in proof of Lemma 4.2: e.g., set up the proof a bit more -- perhaps put line 164 first, then introduce Yi as the indicator of whether the event Xi^2 > c2' holds. It shouldn't read out of order. Lines 191-192 are written confusingly, maybe just write an equation defining W_i^x in terms of ReLU. Be clearer about why sqrt(ln(d)) shows up in the scaling. I didn't follow where O(log(d)) neurons become inactive comes from (line 196). Finally, in lines 205-208, is it the case that the estimate of change on the output of the network is coming from using Lipschitzness? Isn't this backward? e.g. lipschitz says \|\nabla f(x_t)\| = \|x_t+1 - x_t\| \geq (1/L) f(x_t+1) - f(x_t), and we are given a lower bound on the gradient (so how do we bound the change in value of f)? Are you using something about gradient flow? At any rate, more clarification is needed here. Also I mentioned this before, but there needs to be a discussion somewhere of what the output of the network actually is in more formality than line 97. How do the results change if you remove the constant Lipschitz/spectral norm assumptions? It seems like bad Lipschitz constants should help you since there could be vast variability in the function value output for small distances away. Should include full proof of Lemma 4.4 in appendix.


Review 4

Summary and Contributions: This paper aims to show that ReLU networks are vulnerable to adversarial attacks with a small norm. They tackle the problem assuming the weights are randomly normally distributed.

Strengths: The paper tackles an interesting and unsolved problem. - Considering random weights is an interesting approach already used in [1] to show vulnerability, also in the context of random weights - The results are strong and very interesting, if some can assess the correctness of the approach.

Weaknesses: So far, I have many concerns about the current version. - I do not understand theorem 3.1, what is the length of the gradient flow? What does it mean to fool the network with a gradient flow starting at x_0? I admit the result is not clear at all. - In all the paper, I am a bit disturbed by d_k+1=o(d_k), it is not clear for me what it means in this context. - The footnote making equivalence between lower bound on singular values and c-surjectivity does not appear immediate or even false - The proofs are hardly readable: I suggest the authors to write them in a clearer way, it is too difficult to assess the correctness. - There are no experiments backing the theoretical claims: it would be good to illustrate them.

Correctness: As said before, the proofs are hardly readable, the authors could clarify them to make them legible.

Clarity: The paper is hard to follow, I suggest the authors to work again to make it readable. I cannot understand the theorems in their current form.

Relation to Prior Work: [1] uses random weights to show vulnerability of neural nets to adversarial examples. Can the authors compare to them? [1] Adversarial vulnerability of neural networks increases with input dimension. Carl-Johann Simon-Gabriel, Yann Ollivier, Léon Bottou, Bernhard Schölkopf, David Lopez-Paz

Reproducibility: No

Additional Feedback: I suggest the authors to make their paper readable, it might be a very good paper, but in the current form I recommend rejecting the paper, so that the authors can work again on.

[Author Response · NeurIPS 2020]

We thank the reviewers for their efforts. Below we address the main comments.

**Reviewer #1**

We thank the reviewer for his/her positive feedback! Experiments do show that the phenomena described in our results extends to architectures beyond fully connected networks, and deep networks.

We will add details about these experiments in the final version, and leave the extension of our results to more architectures as a future direction.

**Reviewer #2**

We thank the reviewer for his/her very positive feedback!

We will address his comments in the final version of the paper.

**Reviewer #3**

We thank the reviewer for his/her overall positive feedback.

We will address his comments in the final version of the paper.

**Reviewer #4**

We thank the reviewer for his/her feedback. The reviewer is concerned mostly about readability, and we will make every effort to clarify the points that he/she has raised. Yet, given that (1) the reviewer thinks that "The results are strong and very interesting", (2) the positive feedback of the other three reviewers, and (3) the fact that the other reviewers wrote that "the paper is well-rewritten and easy to follow", that "The theoretical claims are sound and the proofs seem correct", and that "(correctness) Seems fine" we ask him/her to reconsider his/her score.

We next address specific concerns:

1. "I do not understand theorem 3.1, what is the length of the gradient flow? What does it mean to fool the network with a gradient flow starting at $x_0$? I admit the result is not clear at all. In all the paper, I am a bit disturbed by $d_{k+1} = o(d_k)$, it is not clear for me what it means in this context."

   The length of gradient flow is the length of the trajectory of gradient flow, until the network sign is flipped.

   "to fool the network with a gradient flow starting at $x_0$" means to start gradient flow from $x_0$ and to reach a point in which the sign of the network is different. That new point is the adversarial example, as it is so close in terms of the Euclidean distance.

   "$d_{k+1} = o(d_k)$" means that we assume that the dimension decreases in every layer. To illustrate that, this holds for instance if $\sqrt{d_k} \leq d_{k+1} \leq d_k/\log(k)$ for every $k$

2. "The footnote making equivalence between lower bound on singular values and c-surjectivity does not appear immediate or even false"

   We will add a proof to the final version

3. "There are no experiments backing the theoretical claims: it would be good to illustrate them."

   Please see our response to reviewer # 1

[Meta-Review · NeurIPS 2020]

This work investigates the phenomenon of adversarial examples for neural networks. It shows that, under certain conditions on the architecture (monotonically decreasing widths of layers), for "most" ReLU networks (with respect to random generation of edge weights), all points permit small distance adversarial perturbations (w.r.t. Euclidean metric) and these points can be found by gradient flow (or gradient descent with sufficiently small step sizes). This study uses techniques from random matrix theory and gradient flows to establish the result. Now experimental validation of the phenomena is proven, which, as some reviewers pointed out, would be useful to have, since the proven results involve potentially large constants (in fact, the work makes heavy use of big-O-notation, which makes the statements and presentation nicely succinct, but can also hide potential issues with relevance of the proven claims). Most reviewers appreciated the results established in this work, providing a sound theoretical explanation for the abundant phenomenon of adversarial examples for DNNs. This phenomenon is currently receiving a lot of research attention in the general ML community, and also of societal interest as neural networks are being employed in a growing range of applications and better understanding of their performance and vulnerabilities is important to develop user trust. This paper brings a new set of techniques to formally understand this phenomenon. Several reviewers have also pointed out weaknesses with regard to the presentation, which the authors are encouraged to improve when preparing their final version. In particular, given that the current manuscript has more than two pages room, the authors should consider: - being more explicit w.r.t. asymptotics, the heavy use of big-Oh notation can be viewed as hiding limitations of the current results, and in some places is also no cleanly used (e.g. Thm, 3.2) - overall, adding more explanations, illustrations, intuition to make the manuscript more accessible to less technically versed readers, flesh out proof sketches etc (consider that neurips is a venue where practitioners and theoreticians in the ML research community come together; personally I appreciate the clean and succinct writing style in the submission, but the authors should make reasonable efforts to make their work accessible to newcomers and the general audience at the venue they aim to publish at; maybe both succinctness and accessibility can be achieved) - adding a small set of experimental illustrations of the proven phenomena The reviews also contain more concrete suggestions along these lines.